# Influence of Anode Immersion Speed on Current and Power in Plasma Electrolytic Polishing

**DOI:** 10.3390/mi15060783

**Published:** 2024-06-14

**Authors:** Joško Valentinčič, Henning Zeidler, Toni Böttger, Marko Jerman

**Affiliations:** 1Faculty of Mechanical Engineering, University of Ljubljana, Aškerčeva 6, 1000 Ljubljana, Slovenia; marko.jerman@fs.uni-lj.si; 2Chair Additive Manufacturing, Technische Universität Bergakademie Freiberg, Agricolastrasse 1, 09599 Freiberg, Germany; henning.zeidler@imkf.tu-freiberg.de (H.Z.); toni.boettger@imkf.tu-freiberg.de (T.B.)

**Keywords:** plasma electrolytic polishing (PeP), vapour–gaseous envelope, immersion speed, peak current, average power, process initialisation

## Abstract

Plasma electrolytic polishing (PeP) is mainly used to improve the surface quality and thus the performance of electrically conductive parts. It is usually used as an anodic process, i.e., the workpiece is positively charged. However, the process is susceptible to high current peaks during the formation of the vapour–gaseous envelope, especially when polishing workpieces with a large surface area. In this study, the influence of the anode immersion speed on the current peaks and the average power during the initialisation of the PeP process is investigated for an anode the size of a microreactor mould insert. Through systematic experimentation and analysis, this work provides insights into the control of the initialisation process by modulating the anode immersion speed. The results clarify the relationship between immersion speed, peak current, and average power and provide a novel approach to improve process efficiency in PeP. The highest peak current and average power occur when the electrolyte splashes over the top of the anode and not, as expected, when the anode touches the electrolyte. By immersion of the anode while the voltage is applied to the anode and counterelectrode, the reduction of both parameters is over 80%.

## 1. Introduction

Plasma electrolytic processes have attracted the attention of the metal finishing industry due to their ability to significantly improve surface properties [1]. In 1950, Kellogg reported the formation of a vapour–gaseous envelope (he named it the “aqueous anode effect”) around an anode and an increased temperature of the anode at voltages up to 120 V [2]. In general, the vapour–gaseous envelope is formed around the smallest electrode. In 1974, Lazarenko et al. acquired voltage signals for voltages up to 300 V and identified the differences between the anodic and cathodic processes [3]. They observed higher voltage oscillation frequency in the anodic process where glow discharges are present and spark discharges occurring only episodically. The current passage through the envelope is explained by the emission of ions from an electrolyte solution under the action of an electric field based on the observed features of the glow spectrum in the heated anode, the possibility of diffusive saturation of the surface layer, and the presence of critical voltages [4].

In the cathodic process, transitions from glow to pulsed-arc discharge occur. The energy released in this area causes additional evaporation of the electrolyte, and the increased pressure expels the electrolyte; the arc discharge is interrupted and arises again in other sections of the electrode. Glow discharge is characterised by relatively small erosion of the cathode, spark discharge by significant erosion of the anode, and arc discharge by cathode erosion, which is much greater than with glow discharge. Since the magnitude of the arc discharge current is significantly greater than that of the glow discharge, the electrode temperature in the cathodic process is always higher than in the anodic process for given values of the applied voltage. Therefore, in the cathodic process, intense heating of the metal up to melting is observed. It is generally accepted that current passage through the envelope is carried out by means of electrical discharges [4].

Therefore, to improve surface roughness, the anodic process is favourable, and most studies are focused on the anodic process, with very few on the cathodic process [5]. The anodic process in electrolytes consists of four modes, described at the end of the 1970s by Duradji et al. [6]. The first mode, observed on the active electrode at low current densities, is typical electrolysis. The resistance is a fixed value, which is the classic electrolysis mode where the Faraday law is valid. The electrolyte directly contacts the surface of the anode. Although bubbles are present on the anode surface, the resistance in the circuit is mainly the resistance of the electrolyte. Increasing the voltage to around 70 V and current density to 15 A cm^−2^ causes the oxygen evolution reaction [7]; numerous bubbles are generated near the anode, bubbles begin to fuse to form a vapour–gaseous envelope, and the resistance value in the circuit rises [8], which leads to the emergence of a transitional or switching mode, characterised by the periodic formation of a vapour–gaseous envelope around the active electrode, causing current interruption for 100 μs. The third mode, a stable one or the “Kellogg mode”, occurs at voltages above 80 V with the formation of a stationary vapour–gaseous envelope around the active electrode and a decrease in current density down to 0.5 A cm^−2^. Electrical conductivity of the envelope is determined by a combination of solution boiling processes and emission of electrolyte anions [4]. Further increasing the voltage after establishing the stable mode intensifies the luminescence of electric discharges, thickens the vapour–gaseous envelope, and in some areas leads to its disruption. This reduces the current by half, establishing a fourth mode with electro-hydrodynamic phenomena in the electrolyte and a different nature of electric discharges within the vapour–gaseous envelope. Zhou et al. [8] report a dynamic stable envelope (the third mode according to Duradji et al.) is formed around 200 V, where the resistance is significantly rising, indicating the rapid thickening of the envelope, which completely separates the anode from the electrolyte and produces a strong glow discharge phenomenon.

Regardless of how the modes are defined, at a certain voltage between two electrodes in an aqueous electrolyte, there is a deviation from Faraday’s normal electrolytic regime [5]. The vapour–gaseous envelope is formed and functions in the film, bubble, or transient boiling modes [9]. The conductivity of the vapour–gaseous envelope that functions in the film type of boiling is caused by electrical microdischarges inside the envelope. In the case of the bubble type of boiling, the conductivity of the envelope is determined by the motion of charges through electrolyte bridges between the bubbles. In the transient boiling mode of the envelope, the bridges evaporate because of the high value of the current; the liquid is pushed away from of the surface, and the current decreases. Next, the springy force of the liquid compresses the vapour–gaseous envelope, and finally, the electrolyte touches the surface. The current increases and the cycle begins again. This kind of the vapour–gaseous envelope behaviour is characterised by periodical oscillations of the current.

Plasma electrolytic polishing (PeP) is a special case of electrochemical machining, where the anodic process is utilised and voltages typically fall between 200 V to 600 V, and thus, material removal is performed utilising electro-hydrodynamic phenomena. The specific voltage used can vary depending on the material being polished, the electrolyte composition, and the desired surface finish. The electrolytes are mostly environmentally friendly, low-concentration aqueous salt solutions [10]. The PeP process, firstly closer examined probably in 1979 [11], is a surface treatment technology that leads to very smooth, high-gloss surfaces with improved corrosion resistance [12] and is a promising postprocessing technology for additively manufactured parts [13].

The process takes place in a vat containing a material-specific aqueous electrolyte solution with low viscosity. Its conductivity is adjusted from 4 S m^−1^ to 30 S m^−1^ by adding up to 12% of various salts and suitable complexing agents. The workpiece in the vat acts as an anode. In addition to the workpiece, there is a cathode electrode in the electrolytic cell. The surface ratio between the anode and cathode should be greater than 1:10, and the cathode does not have to resemble the geometry of the part; a ring or plate shape is common. The ratio is necessary to ensure the required current density for plasma formation on the anode surface. The relationship between current density and applied voltage must be set to adjust the process window to the electro-hydrodynamic area for the PeP process. There, caused by the process conditions, a vapour–gaseous envelope forms around the anode, and a plasma layer is created. In combination with electrochemical reactions such as (anodic) metal dissolution, (anodic) oxide formation, hydrogen formation, and alkalisation, plasma reactions such as ionisation of the vapour–gaseous envelope and hydrothermal reactions such as metal dissolution by metal–water reaction lead to a removal of surface peaks and thus to a polishing of the part. There is significant disagreement in the literature regarding the maximum anode temperature, namely, 120 °C [14], 300 °C [9], and up to 1000 °C [11].

A variety of electrically conductive materials have been successfully polished, namely, a variety of steels [15], copper alloys [14,16,17] including metallic glass [18], aluminium alloys [19], and other materials such as cobalt–chromium alloy [20], titanium alloy [21], nitinol [22], etc. Regardless of which material is to be polished, the current peak and the power that occur during the initialisation phase of the process must be taken into account. When a voltage is applied between the two electrodes submerged in an electrolyte, a high current peak is generated due to the high electrical conductivity of the electrolyte. When the vapour–gaseous envelope forms, the conductivity decreases considerably, and the current drops to several amperes depending on the size of the anode surface [14]. The larger the anode surface, the higher the current and the power.

To avoid high peaks, the anode is immersed in the electrolyte after the voltage is applied; hence, the vapour–gaseous envelope starts to form when the anode touches the electrolyte, and it spreads over the anode as it is immersed in the electrolyte. In the PeP process, the anode temperature plays an important role in forming the envelope. The maximum electrode immersion speed *v* when the heat can be effectively conducted to the nonimmersed part can be determined by Equation (Equation 1) [23]:(1)$$v\leq \frac{\dot{Q}}{c\rho \Delta TA},$$
where $\dot{Q}$ is a heat transfer rate, *c* is a specific heat capacity, ΔT is a temperature difference between the planes at the electrolyte level and the top of the nonimmersed part, and *A* is the anode bottom surface area. Obviously, the equation is valid only for the parts with constant *A*. In this respect, the immersion speed plays a significant role.

During the process initialisation phase, i.e., when the vapour–gaseous envelope is formed, it is advantageous not to exceed the power that is required after the vapour–gaseous envelope has completely formed, i.e., during stable polishing. In extreme situations, the electrical power may be too high, and fuses will switch off the power supply to prevent damage to the power supply and/or electrical wiring; therefore, the process is initiated by slow immersion of an anode into an electrolyte to allow time for the vapour–gaseous envelope to form and to avoid a large surface area being in direct contact with the electrolyte. The influence of the immersion speed on the electrical quantities and phenomena during the initialisation phase has not yet been systematically investigated.

In this paper, the effect of immersion speed (the anode velocity in z direction) on the peak current and average power is investigated in the case of an anode size corresponding to a microreactor baseplate [24] to be manufactured by laser powder bed fusion (PBF-LB/M) and polished by PeP [25]. The problem of initialization of the PeP process is greater when large workpieces are polished. In such cases, the initialization of the process is performed by slow submerging of workpieces. Smaller workpieces, such as microreactor components/tools, can be and often are plasma polished on much smaller machine tools or even test rigs, where the immersion speed cannot be controlled but high power consumption is observed. Hence, the presented research and experimentation explore the initiation of the process at this scale. We formulate the hypothesis that the highest power occurs when the anode comes into contact with an electrolyte and forms a short circuit on a large surface before the vapour–gaseous envelope forms. The peak current occurring there is very important. We also try to determine the highest immersion speed at which the electrical power during process initialisation does not exceed the power required for stable polishing. Based on the results, the hypothesis is rejected but the appropriate immersion speed is successfully defined.

## 2. Materials and Methods

The experiments were carried out with an 80 kVA PeP plant (Leukhardt Schaltanlagen GmbH, Immendingen, Germany), which can continuously supply 150 A and voltages in the range from 150 to 380 V. Besides a rectifier, it has a large capacitor bank (20 mF) that provides enough power to handle short current spikes without damaging the rectifier and transformer. The fuses of the plant are also dimensioned in a way to handle short spikes. The machine has a z stage and enables the control of the anode movement in z direction and thus the setting of the immersion speed in the range from 0 to 500 mm · s^−1^.

The temperature does not depend on the immersion depth in the cathodic process, but in the anodic process, the electrode temperature increases linearly with increasing immersion depth and decreases proportionally with the square root of the electrode radius. The temperature of the immersed part of the electrode is uniform at all points [23]. Therefore, the anode was immersed always to the same depth at various speeds.

The temperature and concentration of the electrolyte play an important role in the PeP process. To create the same conditions in all experiments, sodium carbonate (${Na}_{2}{CO}_{3}$) dissolved in water was used as the electrolyte to avoid material removal on the anode. We kept the temperature of the electrolyte at 80 °C and the concentration of sodium carbonate at 0.55 M. The temperature was controlled by a Caso design TC2400 induction heating hob (Caso Gmbh, Arnsberg, Germany). Due to water evaporation, the electrical conductivity was frequently monitored and maintained at 13.5 S m^−1^; fresh water was added when needed. The anode used for the experiments was the size of a mould insert for the mass production of microreactor units. It was a plate with a diameter of 40 mm and a thickness of 10 mm. The bottom surface was therefore 1256 mm^2^ and the total surface area was 3768 mm^2^. The material was stainless steel AISI 316L, and its bottom surface was mechanically polished to a mirror surface (Ra = 0.05 μm). Thus, a repeatable effective surface area as well as wetting characteristics such as contact angle of the anode were maintained, making the experiments more repeatable and more under control. The bottom anode surface and electrolyte surface were never completely parallel to each other, which is an extraneous parameter in these experiments. A rectangular shape is more sensitive to nonparallelism than a circular shape; hence, the latter was selected despite the fact that the microreactors have a rectangular shape of their bottom surface.

To acquire the voltage and current signals, a Picoscope^®^ 3205D oscilloscope (Pico Technology, Saint Neots, UK) with a resolution of 8 bits at 1 GS s^−1^ and a bandwidth of 100 MHz was connected to a PC (Linux OS with PicoScope^®^ 7 oscilloscope software) via a USB cable. The voltage differential probe Testec TT-SI9101 (Testec Elektronik GmbH, Dreieich, Germany) with a bandwidth of 100 MHz was used with an attenuation ratio of 1:100 to reduce the input voltage to the oscilloscope. The current was measured with a Tektronix TCP303 current probe (Tektronix Inc., Beaverton, OR, USA) with 15 MHz bandwidth and the ability to measure currents up to 150 A. The probe was connected to a Tektronix TCPA300 amplifier (Tektronix Inc., USA) with 100 MHz bandwidth. The entire setup is shown in Figure 1.

The current and voltage waveforms were recorded in an oscilloscope buffer and transferred to the PC after acquisition. They were analysed in a Matlab^®^ R2023b programming environment. Due to the relatively low resolution of the oscilloscope in the *y* axis (8 bits), the voltage and current signals were first filtered using a Savitzky–Golay smoothing filter [26] with second-degree polynomial function, taking into account 0.02 ms before and after the observed sample on the signal (see Appendix A for the effect of filtering). The filtered voltage and current signals were used to calculate the power signal.

Two current peaks were identified in the initialisation phase of the process, namely, when the anode and the electrolyte made electrical contact and when the electrolyte splashed onto the anode’s top surface. Within these two time domains, the average power $\bar{P}$ was calculated as the energy within the time domain divided by the duration of the time domain *t* according to the following equation: (2)$$\bar{P}=\frac{1}{t}\sum_{i=1}^{N}U_{i}\;\cdot \;I_{i}\;\cdot \;\Delta t,$$
where $U_{i}$ is the voltage, $I_{i}$ the current, *i* a sample number, and *N* is the number of samples in the time domain. The average current and power during stable polishing were also calculated for the last 0.5 ms of the signals.

Preliminary experiments were conducted, and the results showed that a significant peak current occurred when the immersion speed is around 100 mm·$s^{-1}$. Hence, the speeds between 100 mm·$s^{-1}$ and 500 mm·$s^{-1}$ were selected linearly with the increment of 100 mm·$s^{-1}$. Since the lower speeds were close to the situation when the anode and electrolyte were in contact during the process initialisation, v=5 mm·$s^{-1}$ and v=20 mm·$s^{-1}$ were examined as well. Therefore, the experiments were carried out at seven immersion speeds, namely, 5, 20, 100, 200, 300, 400, and 500 mm · s^−1^. Two additional experiments were carried out. In the first, the bottom surface of the anode was touching the electrolyte surface, and the contact area was 1256 mm^2^. In the second, the anode was submerged in the electrolyte, and the contact area was 3768 mm^2^. Most of the experiments were performed five times. The exceptions are as follows: experiments with immersion speed 500 mm · s^−1^ were performed three times, immersion speed 400 mm · s^−1^ ten times, and with submerged anode only two times.

A high number of repetitions was performed on a selected immersion speed, i.e., 400 mm·$s^{-1}$, to observe the process repeatability. Based on the findings, a set of five repetitions was selected for all experiments except for the highest immersion speed and for the experiments where the process started with the anode submerged. The highest immersion speed, i.e., 500 mm·$s^{-1}$, is on the edge of the machine tool performance. On the given path in z direction (330 mm), the anode accelerates to the given immersion speed and then decelerates to 0 mm · s^−1^. The immersion speed of 500 mm · s^−1^ was not achieved during some experiments, and those results were eliminated from further processing. The acquired results of the three experiments agree quite well, and thus, only three experiments were conducted at this immersion speed.

When the process starts with the submerged anode, the safety fuses of the machine tool are often activated; hence, it is difficult to perform these experiments successfully. Therefore, the experiments starting with a submerged anode were limited to two repetitions to avoid frequent overloading of the machine tool. The results are highly repeatable (see Appendix A).

## 3. Results and Discussion

The polishing process begins with a short circuit due to the high conductivity of the electrolyte liquid. Once the vapour–gaseous envelope has formed, the resistivity is increased and the current is reduced. When polishing starts with the submerged anode, the peak current is up to 600 A and the peak power is up to 200 kW, as shown in Figure 2. There is also a significant voltage drop until the vapour–gaseous envelope has formed around the entire anode. Hence, even in the case of relatively small workpieces to be polished, like mould inserts for the manufacturing of microreactors, the process should be initiated by immersion of the workpiece and not by starting the process when the workpiece is already submerged in the electrolyte. The three consecutive peaks occur during this initialisation phase. These peaks are present in all acquired signals when the anode is submerged. This high repeatability is also observed in the case when the electrode bottom surface only touches the electrolyte (for repeatability, see Appendix A). Both experiments with a fixed anode position in the z direction show that the formation of vapour–gaseous envelope is a dynamic but relatively repeatable process. The results are in line with observations in [9], where the influence of voltage and electrolyte temperature on the formation of the vapour–gaseous envelope were observed on an cylindrical anode with a diameter of 7 mm and length of 60 mm that was slowly immersed into the electrolyte. A thin, unstable film kind of vapour–gaseous envelope forms around the sample. This kind of the vapour–gaseous envelope is characterised by the fluctuations in the current signal. Constant current sections which correspond to stable film, and three impulses which correspond to destroying the film, can be seen in Figure 2. At the moment of destroying the film, the electrolyte touches the surface of the anode; the solution around it gets warmer because of Joule heat liberation, and the bubbling effect of the electrolyte changes.

To avoid high current peaks during process initialisation, the anode is slowly immersed in the electrolyte. Figure 3 shows the signals acquired when the immersion speed is set to 5 mm · s^−1^. No significant current and power peaks are observed, and the highest current occurs when the electrolyte covers the top anode surface. This event is marked by two vertical red dashed lines.

The anode touches the electrolyte at time 0, where the current appears and a small voltage drop can be observed. The continuous vertical red lines indicate the contact time domain where the peak values are identified (marked by the red circle) and where the average power is calculated. Both the current and the power slowly increase as the anode side (cylindrical) surface is immersed in the electrolyte and the vapour–gaseous envelope spreads over the submerged surface. After the top of the anode reaches the electrolyte surface level, which is indicated by the dotted vertical line at t=2 s, the top surface is splashed over by the electrolyte. After more than half a second, an increase in current (indicated by the first dashed vertical line) can be observed, and about one second later, the complete vapour–gaseous envelope forms around the anode (indicated by the second dashed vertical line). The area between the two dashed lines is used to determine the current peak value and to calculate the average power consumed during the splashing. These values are determined at different immersion speeds: when the anode position was fixed and touching the electrolyte and when it was submerged in the electrolyte.

The current peak level and the time of its occurrence depend on the immersion speed, as shown in Figure 4. A higher immersion speed leads to higher peak values, and the time between the start of polishing and the peak value is shorter. The highest speed the machine tool can deliver is 500 mm · s^−1^. However, even at this immersion speed, the current peaks are ten times lower compared to the current peaks measured when the process starts with a submerged anode or when the anode bottom surface is touching the electrolyte surface. The average current during polishing is calculated based on all acquired signals and displayed as a blue, continuous horizontal line over the entire time span. The dotted lines represent the corresponding standard deviation.

When the anode is immersed, the vapour–gaseous envelope forms before any short circuit effects are observed, preventing significant peak currents from appearing on the waveform. Conversely, when the process starts with the anode already in contact with the electrolyte surface, the entire bottom surface of the anode is wetted by the electrolyte. Upon applying voltage, a high current peak is detected due to the short circuit. The envelope formation time is approximately 3 ms (Appendix A).

According to Equation (Equation 1), the maximum immersion speed at which heat can be effectively conducted to the nonimmersed part is 0.5 mm · s^−1^ (Appendix A). However, this effect is not observed in the acquired waveforms. It can be speculated that when the cold, nonimmersed surface of the anode comes into contact with the electrolyte, the anode heats up and a vapour–gaseous envelope forms much faster than heat is transferred up the anode.

For an anode with flat and parallel top and bottom surfaces, the highest peaks are not observed when the anode comes into contact with the electrolyte but when the electrolyte splashes over the top surface of the anode (Figure 5), which is an interesting result. The top and bottom surfaces of the anode are the same size, and the entire bottom surface immediately causes a short circuit in contact with the electrolyte. Therefore, higher peak values are expected during the contact with the electrolyte and not during the splashing over the anode top surface. However, it appears that the electrolyte flow plays an important role in the formation of the vapour–gaseous envelope at all immersion speeds (Figure 6), and the same is true for the average power (Figure 7). This should be taken into account when planning the PeP process, especially when polishing lattice structures. The report on polishing lattice structures by the PeP process is provided in [22].

It seems the properties of the vapour–gaseous envelope do not vary along the sample and do not depend on the immersion speed. When the anode touches the electrolyte at an immersion speed below 100 mm · s^−1^, the short circuit is instantly transformed into the vapour–gaseous envelope. During splashing, the electrolyte flow causes the highly dynamic process of vapour–gaseous envelope formation that is the most similar to the transient bubble mode. At a higher immersion speed, the short circuit time is longer, which indicates that the increased stagnation pressure in the electrolyte reduces the formation of the vapour–gaseous envelope. Consequently, the current peak is higher and appears faster (Figure 4).

At high immersion speeds ($v\geq 300\;mm\;\cdot \;s^{-1}$), only the bottom surface of the immersed part is in contact with electrolyte until the space that is created by splashing is filled with electrolyte. Both the side and top surfaces get in contact with electrolyte, and thus, a high current peak is noticed during the splashing. In the case of lower immersion speeds ($v\leq 200\;mm\;\cdot \;s^{-1}$), the side surface is in contact with electrolyte during immersion, and the splashing of electrolyte occurs only on the top surface. Hence, the current peak during splashing is not so high. This is clearly seen in the slow-motion videos that are part of the Appendix A (the name of each file denotes the immersion speed, submerged anode, or anode touching the electrolyte surface).

When the anode is immersed into the electrolyte, the oscillation frequency on the current signal is lower than when the splashing of the electrolyte over the anode top surface is completed and stable polishing is established. This is observed at all immersion speeds (Appendix A). Although the frequency changes, the process is always in the electro-hydrodynamic mode, and the vapour–gaseous envelope features the bubble type of boiling during anode immersion and when the anode is submerged.

At velocities below 100 mm · s^−1^, the peak current and average power during stable polishing are lower than these values. Both the current peaks and the average power play a decisive role for the process performance, and they should be considered to avoid overloading the PeP plant. Therefore, immersion speeds below 100 mm · s^−1^ are recommended for the given workpiece.

Even when polishing a small anode with 38 cm^2^ surface area, a very high peak current and average power occurs when the process starts with the submerged anode. The peak current and average power are enormous, (594 A) and 48.9 kW, respectively (Table 1). When the process is initiated at any immersion speed, the highest current peak is identified during splashing (Figure 6 and Figure 7). It is below 80 A, and the highest average power is below 8 kW, which means a reduction of about 90% and above 80%, respectively. The exact values can be found in Table 1, where the peak current and average power for all velocities and both events, namely, contact and splashing, are given. The peak current and average power are compared with the values determined at the highest immersion speed, i.e., $v=500mm\;\cdot \;s^{-1}$. These values are significantly lower at low velocities but much higher when the anode is touching the electrolyte surface during process initialisation (current peak more than six times higher and average power almost three times higher) or when it is submerged in the electrolyte (current peak more than eight times higher and average power almost six times higher). Therefore, the slow formation of the vapour–gaseous envelope is crucial to keep these values as low as possible.

## 4. Conclusions

Although the initialisation of the PeP process is carried out in practise by slow immersion of the anode in the electrolyte, the influence of the speed on the electrical parameters has not yet been investigated. Based on the results presented, the following conclusions can be drawn:When a flat electrode position is fixed in z direction either touching the electrolyte surface or submerged in it, the process initialisation is intense but highly repeatable, indicating that the formation of the vapour–gaseous envelope is a dynamic but relatively repeatable process.The initialisation phase must also be taken into account when polishing relatively small anodes. The process should be initiated by immersion of the anode and not when the anode is submerged in the electrolyte, as current peaks of 600 A and an average power of 50 kW can be reached when polishing an anode with a surface area of 38 cm^2^. If the process is initialised at any immersion speed, the highest values occur in the initialisation phase during splashing: the current peak is below 80 A and the highest average power is below 8 kW, which corresponds to a reduction of about 90% and above 80%, respectively.When using lower velocities, no significant current peaks are observed. In the case of the anode of the size of a mould insert for microreactors, immersion speeds below 100 mm · s^−1^ should be used.In the case of anodes with flat and parallel top and bottom surfaces, the splashing of the anode top surface by electrolyte causes higher current peaks and average power than in the contact of the anode and electrolyte at all velocities examined. This finding rejects the hypothesis that the highest power occurs when the anode comes into contact with an electrolyte due to a short circuit. The electrolyte flow therefore plays an important role in the formation of the vapour–gaseous envelope at all immersion velocities. This finding is important also when developing PeP technology for the polishing of lattice structures.Peak currents during immersion at the highest speed are almost ten times lower than when the anode and electrolyte are in contact before the process starts, with an even greater difference at low immersion speeds. When the anode is immersed, the vapour–gaseous envelope forms before any short circuit effects occur, preventing significant peak currents. Conversely, when the process starts with the anode already in contact with the electrolyte, a high current peak is detected due to the short circuit. The envelope forms in approximately 3 ms, whereas in the case of a submerged anode, it forms in approximately 10 ms.When the anode touches the electrolyte at immersion speeds below 100 mm · s^−1^, the short circuit quickly transforms into a vapour–gaseous envelope. During splashing, the dynamic electrolyte flow creates a transient bubble mode. At higher immersion speeds, longer short circuit times suggest increased stagnation pressure, reducing vapour–gaseous envelope formation, resulting in higher and faster-occurring current peaks.During anode immersion, the current signal’s oscillation frequency is lower than after electrolyte splashing completes and stable polishing resumes across all immersion speeds. Despite frequency changes, the process remains in electro-hydrodynamic mode with bubble-type boiling in the vapour–gaseous envelope during both immersion and submersion.

There likely exists an immersion speed for every shape and size of anode that leads to the current and voltage signal in the initialisation phase comparable to these signals during stable polishing. Initialization of the process by controlled immersion of the anode is one of the solutions to successfully starting the polishing process without overloading the machine tool. There are also other solutions to slowly extend the vapour–gaseous envelope over the anode surface, which are going to be examined.

## Figures and Tables

**Figure 1 micromachines-15-00783-f001:**
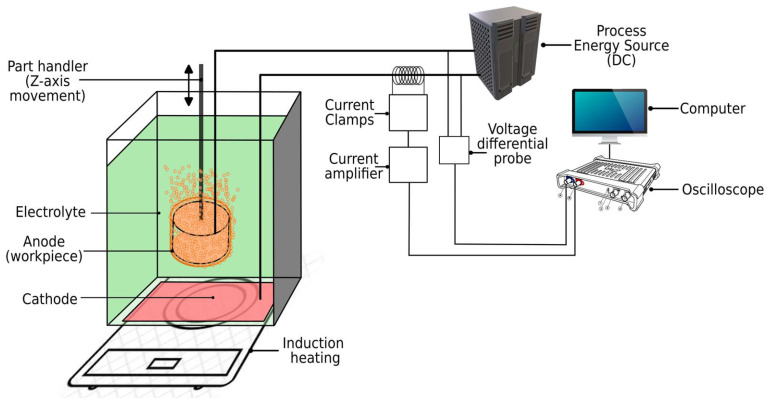
The acquisition system to monitor the voltage and current signals during PeP.

**Figure 2 micromachines-15-00783-f002:**
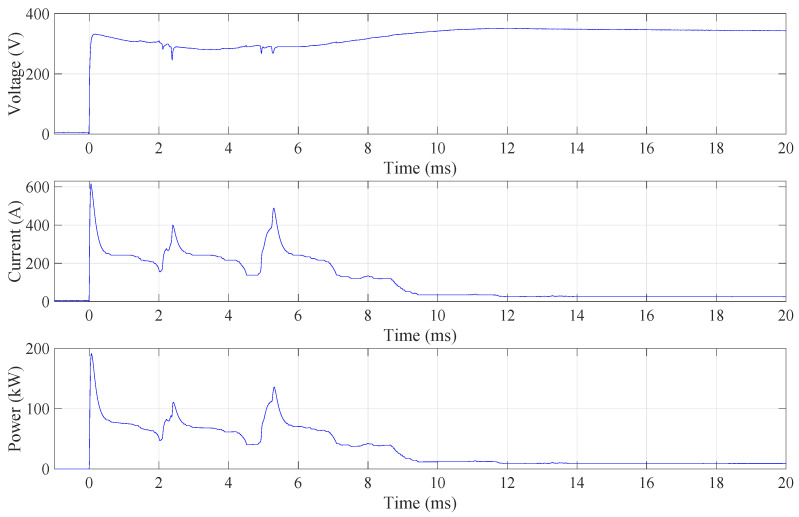
Current and voltage waveform acquired during the process initialisation with submerged anode. The power waveform is calculated in Matlab^®^ programming environment. The initialisation process is highly power-demanding, dynamic, and repeatable.

**Figure 3 micromachines-15-00783-f003:**
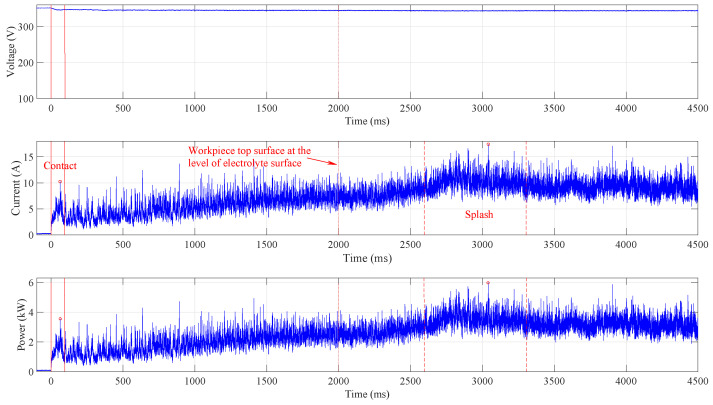
Waveforms acquired and calculated at immersion speed 5 mm · s^−1^. No significant current peaks are identified, and the highest power appears during splashing the electrolyte over the anode top surface.

**Figure 4 micromachines-15-00783-f004:**
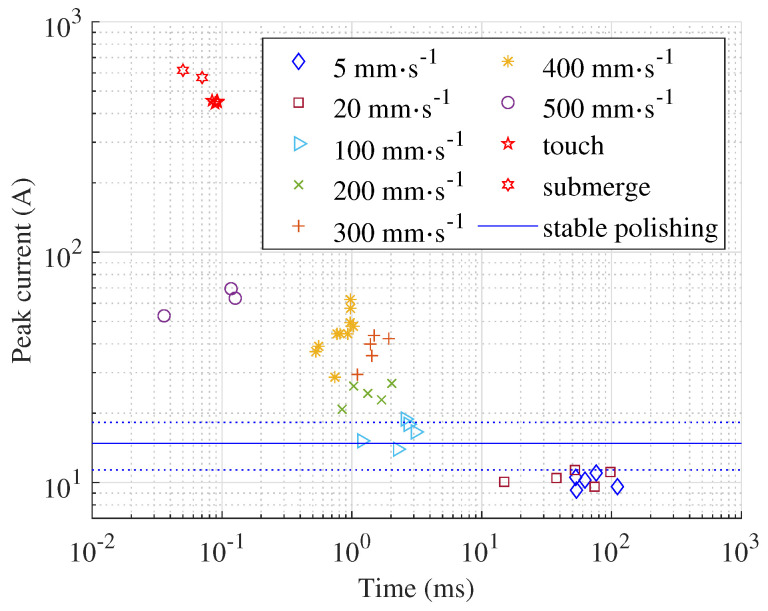
Current peaks in the contact time domain and the time of its appearance measured from the time when anode touches the electrolyte surface for various immersion speeds, when the anode position is fixed at touching the electrolyte surface, and when it is submerged.

**Figure 5 micromachines-15-00783-f005:**
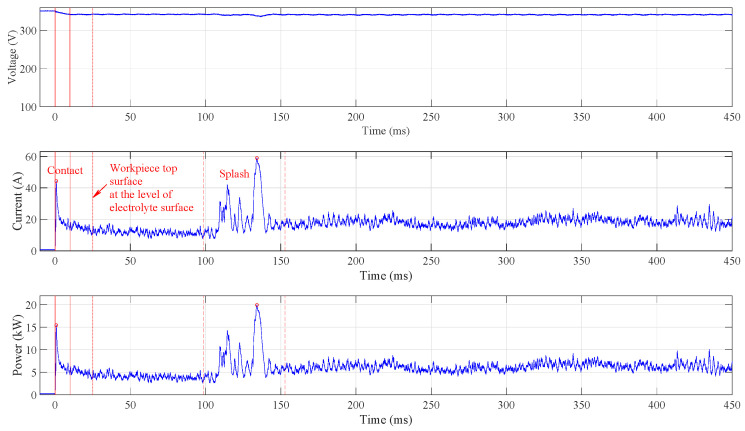
Waveforms at immersion speed 400 mm · s^−1^ and current and power peaks indicated by red circles in contact and splashing time domain.

**Figure 6 micromachines-15-00783-f006:**
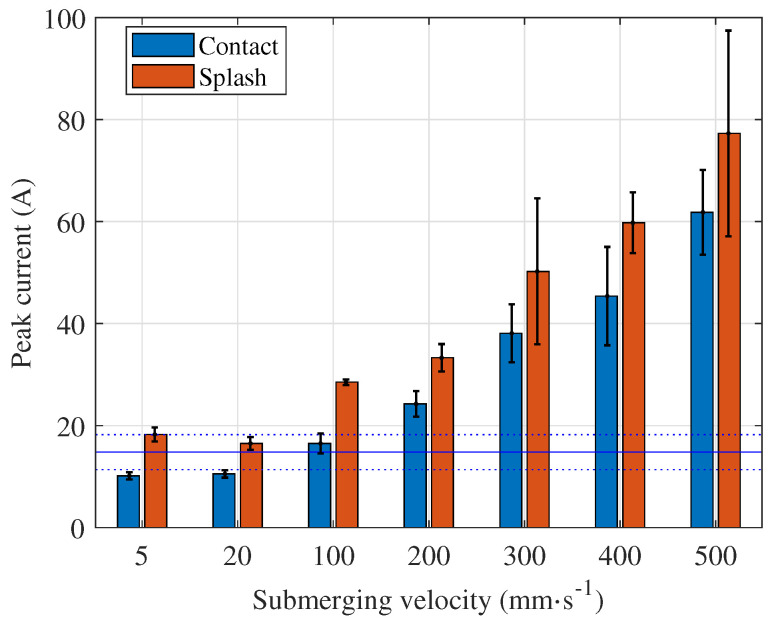
Current peaks for all immersion speeds in contact and splashing time domain. The blue continuous horizontal line represents the average current in stable polishing and the dotted lines its standard deviation.

**Figure 7 micromachines-15-00783-f007:**
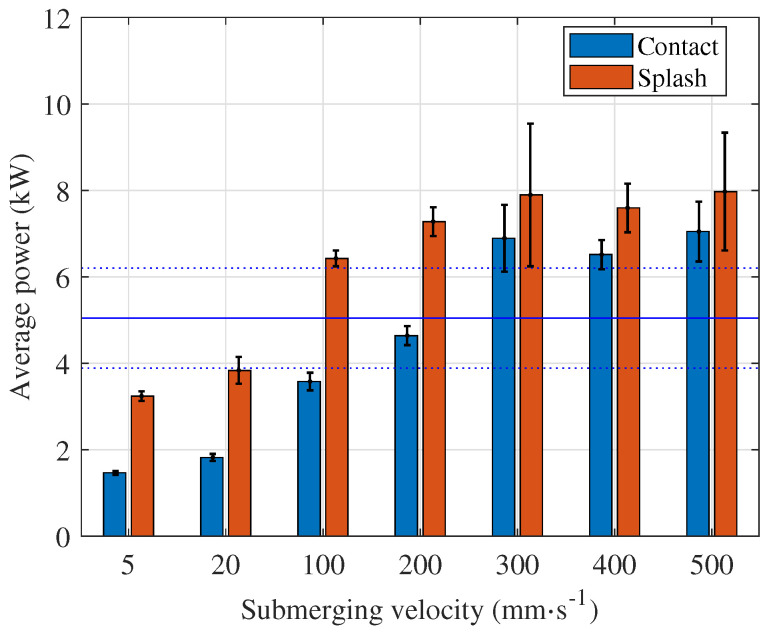
Average power for all immersion speeds in contact and splashing time domain. The blue continuous horizontal line represents the average power in stable polishing and the dotted lines its standard deviation.

**Table 1 micromachines-15-00783-t001:** Values of peak current and average power for all immersion speeds, when the anode position is fixed at touching the electrolyte surface, and when it is submerged. The difference relative to values at the immersion speed 500 mm s^−1^ is given in percentage.

	Contact				Splash			
Immersion Speed (mm $s^{-1}$)	Peak Current (A)	Compared to $500\;\mathrm{mm}\;s^{-1}$	Average Power (kW)	Compared to $500\;\mathrm{mm}\;s^{-1}$	Peak Current (A)	Compared to $500\;\mathrm{mm}\;s^{-1}$	Average Power (kW)	Compared to $500\;\mathrm{mm}\;s^{-1}$
5	10.1	−84%	1.5	−79%	18.3	−76%	3.2	−60%
20	10.5	−83%	1.8	−74%	16.5	−79%	3.8	−53%
100	16.5	−73%	3.6	−49%	28.5	−63%	6.4	−20%
200	24.2	−61%	4.6	−34%	33.3	−57%	7.3	−9%
300	38.1	−38%	6.9	−1%	50.2	−35%	7.9	−1%
400	45.4	−27%	6.5	−7%	59.8	−23%	7.6	−5%
500	61.8		7.0		77.3		7.0	
Touching *	449.4	627%	26.8	283%				
Submerged *	594.0	861%	48.9	599%				

* The anode position was fixed.

## Data Availability

The original contributions presented in the study are included in the article/Appendix A and the data is available in Zenodo (10.5281/zenodo.11637116). Further inquiries can be directed to the corresponding author.

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
