# Peer review of "Influence of Anode Immersion Speed on Current and Power in Plasma Electrolytic Polishing"

_micromachines, 2024, doi:10.3390/mi15060783_

Round 1
Reviewer 1 Report
Comments and Suggestions for Authors
The research content of this article is relatively simple, and its scientific value needs to be improved. The importance of scientific issues is not expressed clearly, and the conclusions drawn are based on common knowledge. I suggest reviewing the manuscript again after the major revision.
In abstract, "However, the process is susceptible to high current peaks during the formation of the vapour skin especially when polishing workpieces with a large surface area."Why is the research subject microreactors? There are significant differences in geometric dimensions.
Author Response
Thank you for considering the paper for publication in Micromachines and for the comments and suggestions, which contributed to higher quality of the manuscript. Your comments (black) have been addressed and we hope the quality of the paper is on sufficient level, now. The major improvements in the manuscript and Supplementary file are written in blue. After considering the comments from the reviewers, the “immersion” is a better expression than “submersion”, hence the “submerging velocity” is changed into “immersion speed” and “workpiece” is renamed into “anode”, also in the title of the manuscript.
The research content of this article is relatively simple, and its scientific value needs to be improved. The importance of scientific issues is not expressed clearly, and the conclusions drawn are based on common knowledge. I suggest reviewing the manuscript again after the major revision.
The importance of scientific issues is now more emphasized throughout the manuscript.
In abstract, "However, the process is susceptible to high current peaks during the formation of the vapour skin especially when polishing workpieces with a large surface area."Why is the research subject microreactors? There are significant differences in geometric dimensions.
The problem of initialization of the plasma electrolytic polishing (PeP) process is greater when large workpieces are polished. In such cases, the initialization of the process is done by slow submerging of workpieces. A smaller workpieces, such as microreactor components/tools, can be and often are plasma polished on much smaller machine tools or even test rigs, where the velocity in z-axis is not controlled, but high power consumption is observed. Hence, the presented research and experimentation explores the initiation of the process at this scale.
The bottom workpiece surface and electrolyte surface are never completely parallel to each other, which is a extraneous parameter in these experiments. Rectangular shape is more sensitive to nonparallelism than circular shape, hence the latter was selected despite the fact that the microreactor mentioned in the literature has rectangular shape of the bottom surface.
We have been using two dimensions of microreactors, namely 35x451 mm (1575 mm2) and 30x30 mm2 (900 mm2). The latter is mentioned in the manuscript where the workpiece of 1256 mm2 (d=40 mm) was used and it represents a typical size of microreactor unit.
The explanation is added in the Materials and methods.
1Sabotin, I., Tristo, G., Junkar, M., & Valentinčič, J. (2013). Two-step design protocol for patterned groove micromixers. Chemical Engineering Research and Design, 91(5), 778–788. https://doi.org/10.1016/j.cherd.2012.09.013
2Sabotin, I., Jerman, M., Lebar, A., Valentinčič, J., Böttger, T., Kühnel, L., & Zeidler, H. (2022). Effects of plasma electrolytic polishing on SLM printed microfluidic platform. Advanced Technologies & Materials, 47(1), 19–23. https://doi.org/10.24867/ATM-2022-1-004
Reviewer 2 Report
Comments and Suggestions for Authors
The article presents an important research direction on reducing the current and power values during the initiation of the plasma electrolytic polishing process. The proposed solution can significantly improve the efficiency and characteristics of the POP process.
Despite a relatively detailed description of the individual stages of research and the results obtained, there is some dissatisfaction with the lack of an in-depth analysis of physical phenomena to the observed results.
Please comment on whether the properties of the vapor gas envelope may vary along the sample and also depend on the immersion speed.
What is physically the main reason for the change in current intensity depending on the immersion speed, and what results in the lack of a significant increase in the current value during contact of the electrode with the electrolyte?
Please comment on why the tests were repeated 10 times at the immersion speed of 400 mms-1 and 3 times at the subdivision speed of 500 mms-1.
Why were full immersion tests performed only twice? Can the results from these tests be considered repeatable?
There are also minor editorial errors in the work that should be corrected.
On line 116 in the sentence: The experiments were carried out at seven submerging velocities, namely 5, 20, 100, 115 200, 300, 400, and 500mms−2., same as line 200, 225, please correct the unit, it should be mms−1.
Author Response
Thank you for considering the paper for publication in Micromachines and for the comments and suggestions, which contributed to higher quality of the manuscript. Your comments (black) have been addressed and we hope the quality of the paper is on sufficient level, now. The major improvements in the manuscript and Supplementary file are written in blue. After considering the comments from the reviewers, the “immersion” is a better expression than “submersion”, hence the “submerging velocity” is changed into “immersion speed” and “workpiece” is renamed into “anode”, also in the title of the manuscript.
Despite a relatively detailed description of the individual stages of research and the results obtained, there is some dissatisfaction with the lack of an in-depth analysis of physical phenomena to the observed results.
The in-depth analysis of physical phenomena is provided in the manuscript in the Introduction and in Results and discussion. The description of the phenomena is based on the pioneering papers on plasma electrolytic processes, mostly in Russian language, but recent findings are provided as well.
Please comment on whether the properties of the vapor gas envelope may vary along the sample and also depend on the immersion speed.
It does not seem that the properties of the vapour-gaseous envelope vary along the sample and the properties do not depend on the immersion speed. When the anode touches the electrolyte at any immersion speed, the short circuit is established immediately the vapour-gaseous envelope is formed. During splashing, the electrolyte flow causes highly dynamic process of vapour-gaseous envelope formation that is the most similar to transient bubble mode. But we think we should not relate to this or any other mode (film, bubble, transient boiling), since the flow of the electrolyte is not stimulated only by the plasma electrolytic process. A thorough research plan is needed in order to make firm conclusions.
When the anode is immersing into the electrolyte, the oscillation frequency on the current signal is lower than when the splashing of the electrolyte over the anode top surface is completed and a stable polishing is established. This is observed at all immersion speeds. The acquired signals are now presented in Supplementary file and a paragraph is added in the manuscript in section Results and discussion. Although the frequency changes, the process is always in the electro-hydrodynamic mode and the vapour-gaseous envelope features bubble type of boiling during the anode immersion and when the anode is submerged.
A few paragraphs are added in the Results and discussion as well as in Conclusions
What is physically the main reason for the change in current intensity depending on the immersion speed, and what results in the lack of a significant increase in the current value during contact of the electrode with the electrolyte?
The formation of vapour-gaseous envelope takes around 3 ms when the anode is at the fixed position touching the electrolyte (now, the waveforms are presented in Supplement file section S5) and 10 ms when the process is initiated with submerged anode. In case the voltage is on when the anode is immersed at speeds lower that 100 mm s-1 in the electrolyte, the vapour-gaseous envelope forms before the effect of short circuit is noticed. Hence, there is no significant peak current detected on the current waveform. At higher immersion speeds, the short circuit time is longer before the vapour-gaseous envelope is formed due to the presence of stagnation pressure in the electrolyte. Consequently current peak is higher and appears faster (Fig. 4).
This explanation is added in the Results and discussion as well as in Conclusions
Please comment on why the tests were repeated 10 times at the immersion speed of 400 mms-1 and 3 times at the subdivision speed of 500 mms-1.
First of all, you are right, “immersion” is a better term than “submerging”. This is corrected throughout the whole manuscript.
The high number of repetitions was performed on selected submerging velocity, i.e. 400 mm s-1 to observe the process repeatability. Based on findings, five repetitions were selected for all experiments except for the highest workpiece velocity and for the fully submerged workpiece.
The highest workpiece velocity is at the edge of the machine tool performance. On the given path in z-direction (all together less than half a metre), the workpiece accelerates to the given velocity, in this case 500 m s-1, and then decelerates to 0 mm s-1. This velocity was not always achieved during experiments and those results were eliminated from further processing. The acquired results of the three experiments agree quite well and, according to our opinion, the three repetitions are enough to confirm the findings.
Two short paragraph are added in the manuscript, but maybe it does not make sense to provide the last one in the manuscript.
Why were full immersion tests performed only twice? Can the results from these tests be considered repeatable?
In the case of fully submerged workpiece, the safety fuses of the machine tool were often activated, hence it is difficult to perform experiments successfully. For “submerged” experiments we decided to limit the repetitions to two, to avoid overloading of the machine tool. Yes, the results are repeatable, since no additional movement of the electrolyte is initiated. In the supplement file, section S2, the signals of both experiments are presented and they are almost identical.
A short paragraph is added in the section Materials and methods.
There are also minor editorial errors in the work that should be corrected.
On line 116 in the sentence: The experiments were carried out at seven submerging velocities, namely 5, 20, 100, 115 200, 300, 400, and 500mms−2., same as line 200, 225, please correct the unit, it should be mms−1.
These and some other editorial errors are corrected, but not written in blue in the manuscript.
Reviewer 3 Report
Comments and Suggestions for Authors
The research topic is very interesting and the results could be useful to better understand and define industrial parameters of the EPP process.
The approach presented in the article , however, is too weak and not sufficiently rigorous from a scientific point of view; in this form it seems like a typical optimization procedure, the effects are described in a too simplistic way and the conclusions sound correct because they are based on reasonable evidences.
I ask the authors to try to make a deep general revision considering these points (please clarify or discuss):
-The polarity of the process (described as anodic and negatively charged).
-The PeP rectifier is 80kW with max 150A and 380V; in touched or submerged position the current reach 450 A and 600 A; how is possible ? please give more informations on the model you used and if possible how manage spikes overload and and short-circuite protection.
- Please give more informations on used sampling rate, data collecting, averaging and the filtering and smoothing of this particular kind of signals. I suggest to present all the data in the supplementary file. Why you didn't do it ?
-why do you used this particular geometry (a cylinder and not a plate also submerged vertically) I understand that you have this kind of sample, but ...
-How dio you explain that in the "touched" configuration the peak is so high when compared with the ones at low velocity._
-please consider the contribution of the immersed part when considering the splashing peak.
-How dio you explain that you have always three peeks with submerged samples(depending on fitting or not) ?
- If possible please specify the reason of this particular set of velocities and not linear or quadratic or....
I found that one of the authors (Henning Zeidler ) published similar investigations; I appreciated the article and suggest to consider that article as example of required level.
My best,
Author Response
Thank you for considering the paper for publication in Micromachines and for the comments and suggestions, which contributed to higher quality of the manuscript. Your comments (black) have been addressed and we hope the quality of the paper is on sufficient level, now. The major improvements in the manuscript and Supplementary file are written in blue. After considering the comments from the reviewers, the “immersion” is a better expression than “submersion”, hence the “submerging velocity” is changed into “immersion speed” and “workpiece” is renamed into “anode”, also in the title of the manuscript.
The approach presented in the article , however, is too weak and not sufficiently rigorous from a scientific point of view; in this form it seems like a typical optimization procedure, the effects are described in a too simplistic way and the conclusions sound correct because they are based on reasonable evidences.
I ask the authors to try to make a deep general revision considering these points (please clarify or discuss):
-The polarity of the process (described as anodic and negatively charged).
The anodic and cathodic processes are describe in the Introduction. Briefly, in anodic process, higher voltage oscillation frequency is observed. Glow discharges are present and spark discharges occurring only episodically, whereas in cathodic process, transitions from glow to pulsed-arc discharge occurs. Hence, higher temperature and material removal rate is observed in cathodic process (Lazarenko 1974).
-The PeP rectifier is 80kW with max 150A and 380V; in touched or submerged position the current reach 450 A and 600 A; how is possible ? please give more informations on the model you used and if possible how manage spikes overload and and short-circuite protection.
The machine tool is not only a 80 kVA rectifier. It’s a complete plant that can provide 80 kVA continuous power. Beside of a rectifier it has a large capacitor bank (20 mF) that provides enough power to handle the short current-spikes without damaging the rectifier and transformer. The fuses of the plant are also dimensioned in a way to handle the short spikes.
A short description is added to Materials and methods.
- Please give more informations on used sampling rate, data collecting, averaging and the filtering and smoothing of this particular kind of signals. I suggest to present all the data in the supplementary file. Why you didn't do it ?
All the data will be published on Zenodo, hence this information was not added to the Supplementary file. Now it is added as S1 and the rest of the sections were renumbered accordingly.
-why do you used this particular geometry (a cylinder and not a plate also submerged vertically) I understand that you have this kind of sample, but ...
A similar observation was given by the first reviewer and the reply here is quite similar as it was given to him/her:
The problem of initialization of the plasma electrolytic polishing (PeP) process is greater when large workpieces are polished. In such cases, the initialization of the process is done by slow submerging of workpieces. A smaller workpieces, such as microreactor components/tools, can be and often are plasma polished on much smaller machine tools or even test rigs, where the velocity in z-axis is not controlled, but high power consumption is observed. Hence, the presented research and experimentation explores the initiation of the process at this scale.
The bottom workpiece surface and electrolyte surface are never completely parallel to each other, which is a extraneous parameter in these experiments. Rectangular shape is more sensitive to nonparallelism than circular shape, hence the latter was selected despite the fact that the microreactor mentioned in the literature has a rectangular shape of the bottom surface.
The cylindrical sample was carefully prepared. The bottom surface was polished to mirror quality to obtain the best contact with electrolyte and hence the highest current peaks. Despite this setup, the peak current during splashing is higher than when the contact between the workpiece and the electrolyte is established.
An explanation is added in the section Materials and methids.
-How dio you explain that in the "touched" configuration the peak is so high when compared with the ones at low velocity.
When the anode is in contact with the electrolyte before the process is initiated, whole surface area is wetted by the electrolyte and thus in a short circuit. When voltage is applied, the high current peak is observed and then the vapour-gaseous envelope is formed, and hence the current is reduced. Regarding the immersion with the voltage on, the answer is the similar as to Reviewer 2. The formation of vapour-gaseous envelope takes around 3 ms when the anode is at the fixed position touching the electrolyte (now, the waveforms are presented in Supplement material section S5) and 10 ms when the process is initiated with submerged anode. In case the voltage is on when the anode is immersed at speeds lower that 100 mm s-1 in the electrolyte, the vapour-gaseous envelope forms before the effect of short circuit is noticed. Hence, there is no significant peak current detected on the current waveform.
This explanation is added in the Results and discussion as well as in Conclusions
-please consider the contribution of the immersed part when considering the splashing peak.
At high immersion speeds (v ≥ 300 mm s-1), only the bottom surface of the immersed part is in contact with electrolyte until the space that is created by splashing is filled with electrolyte. Only now the side and top surfaces are in contact with electrolyte and thus a high current peak is noticed during the splash. In the case of lower immersion speeds (v ≤ 200 mm s-1), the side surface is in contact with electrolyte during immersion and splash of electrolyte occurs only on the top surface. Hence, the current peak during splashing is not so high. The Supplement material is enriched with videos showing the process initialisation at all speeds, submerged anode, and when the anode is touching the electrolyte.
A paragraph is added in the section Results and discussion.
-How dio you explain that you have always three peeks with submerged samples(depending on fitting or not) ?
Such waveforms were observed by Nevyantseva et al. (2001) and explained as follows. A thin unstable film kind of vapour-gaseous envelope forms around the sample. This kind of the vapour-gaseous envelope is characterized by the fluctuations of current signal. Constant current sections which correspond to stable film, and three impulses which correspond to destroying the film. At the moment of destroying the film, the electrolyte touches the surface of the anode; the solution around it gets warmer because of Joule heat liberation, and the bubbling effect of the electrolyte changes.
A paragraph in Results and discussion is extended with this information.
- If possible please specify the reason of this particular set of velocities and not linear or quadratic or....
Yes, there is a reason behind. The machine tool allows the immersion speeds to be in the range of 1 mm s-1 to 500 mm s-1. Speeds above 100 mm s-1 are resulting in the occurrence of current peak and it is interesting to examine the process initialisation in this range of speeds. Hence, the speeds are distributed linearly in this range. The lower speeds are close to situation when the workpiece and electrolyte are in contact (touch) when the voltage is applied. Therefore, the 5 mm s-1 and 20 mm s-1 were selected.
A few sentences are added in the manuscript.
I found that one of the authors (Henning Zeidler ) published similar investigations; I appreciated the article and suggest to consider that article as example of required level.
The manuscript is improved also in this manner.
Round 2
Reviewer 3 Report
Comments and Suggestions for Authors
I appreciated the new version modified also considering the suggested changes. The level has certainly improved and the article is suitable for publication.
best regards,